# Bone Marrow Stroma-Induced Transcriptome and Regulome Signatures of Multiple Myeloma

**DOI:** 10.3390/cancers14040927

**Published:** 2022-02-13

**Authors:** Sebastian A. Dziadowicz, Lei Wang, Halima Akhter, Drake Aesoph, Tulika Sharma, Donald A. Adjeroh, Lori A. Hazlehurst, Gangqing Hu

**Affiliations:** 1Department of Microbiology, Immunology & Cell Biology, West Virginia University, Morgantown, WV 26505, USA; sedziadowicz@mix.wvu.edu (S.A.D.); lei.wang1@hsc.wvu.edu (L.W.); ha00014@mix.wvu.edu (H.A.); da00009@mix.wvu.edu (D.A.); tulika.sharma@hsc.wvu.edu (T.S.); 2Lane Department of Computer Science & Electrical Engineering, West Virginia University, Morgantown, WV 26506, USA; donald.adjeroh@mail.wvu.edu; 3WVU Cancer Institute, West Virginia University, Morgantown, WV 26506, USA; lahazlehurst@hsc.wvu.edu; 4Department of Pharmaceutical Sciences, School of Pharmacy, West Virginia University, Morganton, WV 26506, USA

**Keywords:** multiple myeloma, bone marrow stromal cells, de novo drug resistance, transcriptome, regulome, epigenetic regulation, pioneer factors, JUNB, ATF4::CEBPβ, transcriptome and regulome signatures

## Abstract

**Simple Summary:**

The bone marrow (BM) microenvironment provides a protective sanctuary for multiple myeloma (MM) against therapeutic agents. MM cells interact with BM stromal cells (BMSCs) and the interaction is sufficient to confer de novo multi-drug resistance with epigenetic mechanisms as one of the contributors yet to be elucidated. We profiled genome-wide landscapes of gene expression (transcriptome) and chromatin accessibility (regulome) for MM cells interacting with BMSCs and characterized the induced signatures. We evaluated the contributions from soluble factors derived from BMSCs and compared these results to physical adhesion to the BMSC-induced changes in the transcriptome and regulome. The multi-omics approach further identified candidate transcription factors that regulate the BMSC-induced transcriptome through modulating the regulome, which may lead to promising novel therapeutic targets for the treatment of MM.

**Abstract:**

Multiple myeloma (MM) is a hematological cancer with inevitable drug resistance. MM cells interacting with bone marrow stromal cells (BMSCs) undergo substantial changes in the transcriptome and develop de novo multi-drug resistance. As a critical component in transcriptional regulation, how the chromatin landscape is transformed in MM cells exposed to BMSCs and contributes to the transcriptional response to BMSCs remains elusive. We profiled the transcriptome and regulome for MM cells using a transwell coculture system with BMSCs. The transcriptome and regulome of MM cells from the upper transwell resembled MM cells that coexisted with BMSCs from the lower chamber but were distinctive to monoculture. BMSC-induced genes were enriched in the JAK2/STAT3 signaling pathway, unfolded protein stress, signatures of early plasma cells, and response to proteasome inhibitors. Genes with increasing accessibility at multiple regulatory sites were preferentially induced by BMSCs; these genes were enriched in functions linked to responses to drugs and unfavorable clinic outcomes. We proposed JUNB and ATF4::CEBPβ as candidate transcription factors (TFs) that modulate the BMSC-induced transformation of the regulome linked to the transcriptional response. Together, we characterized the BMSC-induced transcriptome and regulome signatures of MM cells to facilitate research on epigenetic mechanisms of BMSC-induced multi-drug resistance in MM.

## 1. Introduction

There was an estimated 34,920 new cases of Multiple Myeloma (MM) and 12,410 deaths in the United States for 2021 [1]. Despite novel therapeutic options, the inevitable emergence of drug resistance demands improved knowledge of the contributing mechanisms. Bone marrow (BM) constitutes a sanctuary for MM to survive, proliferate, and progress [2,3,4]. The malignant plasma cells interact with cellular components of bone marrow, such as bone marrow stromal cells (BMSCs), and non-cellular components, such as soluble factors and extracellular matrices; the interaction is sufficient to cause drug resistance independent from genetic mutations [2,3,4,5]. This de novo drug resistance protects MM from chemotherapy, contributes to minimal residual disease (MRD), and increases the chance of acquired drug resistance [6]. Thus, investigating the mechanisms of BM-mediated drug resistance helps to define new therapeutic targets to pursue for drug development. The goal of this development is to increase the efficacy of combination strategies and delay the emergence of drug resistance associated with standard-of-care agents used to treat MM patients.

MM cells interacting with BMSCs activate various signaling pathways, including the well-studied JAK2/STAT3, PI3K/AKT, and NFκB [7,8] pathways. The incorporation of external signals from the BM microenvironment into the transcription response of signaling pathways requires cooperative actions of responsive transcription factors (TFs) and chromatin regulators acting on their downstream targets [8,9]. In contrast to acquired drug resistance, de novo drug resistance occurs quickly and is reversible, suggesting that epigenetic mechanisms assume a critical role in regulating the transcription program underlying the resistance [3]. Recent literature has documented several chromatin regulators in drug resistance and myelomagenesis [9,10,11,12,13,14,15], including in the context of the BM microenvironment [5,16]. However, few have investigated the contribution of chromatin reorganization to the BMSC-induced transcription changes in MM cells and the associated regulatory mechanisms linked to de novo drug resistance.

To this end, we profiled, analyzed, and integrated the transcriptome and regulome of MM cells using an in vitro transwell coculture system, where MM cells resided in both the upper transwell (exposed to soluble factors) and the lower chamber where MM cells were exposed to both soluble factors as well as direct contact with BMSCs. These results revealed that the BMSC-induced transformation of the transcriptome and regulome in MM cells was mainly driven by soluble factors. We characterized the transcriptome signatures by aligning to the gene signatures of early plasma cells, Pre-PC cells, and drug response to proteasome inhibitors (PIs). Multi-omics analysis of the changes in the regulome and transcriptome identified candidate TFs that modulate accessible chromatin and contribute to the transcription program linked to BMSC-induced drug resistance.

## 2. Materials and Methods

### 2.1. Cell Lines

MM cell lines MM.1S (ATCC, CRL-2974) and RPMI8226 (ATCC, CCL-155) and the BM stromal cell line HS5 (ATCC, CRL-11882) were sourced from the American Type Culture Collection (Manassas, VA, USA). Cell lines were validated by the short tandem repeat (STR) and tested for mycoplasma every six months. Cells were cultured in RPMI-1640 complete media with L-Glutamine (Corning, Waltham, MA, USA, 10-040-CV) supplemented with 10% Fetal Bovine Serum (FBS) (Gibco, Waltham, MA, USA, 10082-147) and 1% penicillin/streptomycin (Gibco, Waltham, MA, USA, 15140122) (growth media). Cells were grown in a Heracell™ VIOS 160i CO_2_ Incubator with 5% CO_2_ at 37 °C.

### 2.2. Cell Culture

We seeded 2.5 × 10^5^ HS5 cells into the bottom of a six-well transwell plate with 2 mL growth media and left them overnight. The following day, media were removed, and 2.5 × 10^5^ MM cells were added to the stromal layer with 2 mL of growth media. At this time, the transwell was introduced into the system, and 2.5 × 10^5^ multiple myeloma cells were added to the upper transwell with 2 mL of growth media (Thermo Sci, Waltham, MA, USA, 140640). For the monoculture, 2.5 × 10^5^ MM cells were seeded into the bottom of a six-well plate with 2 mL of growth media.

### 2.3. Cell Isolation

Following 48 h incubation, cells were collected in a 15 mL tube. For co-culture conditions, cells were washed with PBS, 1 mL of 0.25% Trypsin (Gibco, Waltham, MA, USA, 15050065) was added, and they were incubated at 37 °C for 5 min. Five milliliters of growth media were added to neutralize the trypsin. Cells were then centrifuged and resuspended in 300 µL FACS staining buffer (1× PBS, 1%FBS). Three microliters of human IgG (Sigma, St. Louis, MO, USA, I-4506) were added to each tube for blocking, and the tubes were incubated on ice for 30 min. Cells were pelleted by centrifugation (500× *g*, 5 min, 4 °C) and the supernatant was removed. Cells were then resuspended in 50 µL of FACS buffer containing 0.5 uL of live/dead stain and 2.5 µL of the appropriate fluorochrome-tagged antibody and incubated on ice for 30 min. The PE anti-human CD38 Antibody (356604) and FITC anti-human CD90 (Thy1) Antibody (38108) were sourced from Biolegend (San Diego, CA, USA). The LIVE/DEAD™ Fixable Near-IR Dead Cell Stain was sourced from Invitrogen (Waltham, MA, USA, L10119A). Following incubation, cells were pelleted, the supernatant was removed, and the pellet was washed twice with 100 µL FACS buffer. Finally, the pellet was resuspended in 300 µL of FACS buffer and flow sorted with a Becton-Dickinson FACS-ARIA II for CD90^−^CD38^+^ cells.

### 2.4. RNA-Seq

RNA was isolated from approximately 10^5^ sorted cells for each sample using the Qiagen kit (RNeasy Plus Mini Kit, 74134) according to the manufacturer’s instructions. RNA was quantified using nanodrop/qubit and RNA integrity and was assessed through the Agilent Bioanalyzer 2100 (Agilent technologies, Santa Clara, CA, USA). Furthermore, 250–500 ng of total RNA was used for each sample as a starting material for RNA-Seq library prep by the WVU Genomics Core using the KAPA mRNA HyperPrep Kit (Roche Diagnostics, Wilmington, MA, USA, 08098123702) according to the manufacturer’s instruction. Libraries were sequenced by the Genomics Core Facility at Marshall University with Illumina Nextseq2000.

### 2.5. Omni-ATAC

Omni-ATAC libraries were prepared with approximately 5 × 10^4^ sorted cells for each using a Tagment DNA Enzyme and Buffer Large Kit (Illumina, San Diego, CA, USA, 20034198) and following the Omni-ATAC protocol [17]. The ATAC libraries were run on 2% agarose gel to observe the nucleosome phasing pattern and size select. For each sample, the gel was cut for an approximately ~200–600 bp band and subjected to gel elution using the Qiagen gel MinElute Gel Extraction Kit (Qiagen, Hilden, Germany, 28606). The libraries were quantified using nanodrop. The quality was assessed using the Agilent bioanalyzer 2100. Libraries were sequenced at the Genomics Core Facility at Marshall University with Illumina Nextseq2000.

### 2.6. Data Analysis

#### 2.6.1. RNA-Seq Data Analysis

RNA-Seq data analysis followed our previous work [18]. Briefly, alignment of paired-end short reads to the reference human genome (hg38) was performed with subread v2.0.1 [19]. Reads aligned to RefSeq transcripts were summarized at the gene level by feature counts [20]. The gene expression level was measured by RPKM (reads per kilobase of exon model per million mapped) [21]. Differentially expressed (DE) genes were predicted by EdgeR3 [22] with FDR < 0.05, fold change > 1.5, and CPM (count per million; log_2_) > 0. The prediction of potential transcription regulators from DE genes was performed by TFEA.ChIP [23].

Gene Set Enrichment Analysis (GSEA) [24] was used for functional inference by evaluating the significance of a preferential expression upregulation in one condition over another for a reference gene set. For each comparison, we created a so-called “rnk” file with the gene symbol in the first column and the fold change of expression in the second column. The rnk file was supplied as the input to GSEA (v4.0.2) with reference sets including the MSigDB hallmark gene set collection [25], signature genes of early plasma cells from the NB2PC atlas (http://www.genomicscape.com/; accessed on 14 January 2021), Pre-PC signature [26], cytokine-induced genes [27], IL6-induced genes [28,29], and adhesion-induced genes [5]. We also used GSEA to validate the signature genes defined in our work, this time treated as the reference set. The rnk file was then generated from external expression datasets that quantify the expression changes in MM.1S in coculture with HS5 as relative to the monoculture [30,31].

We noted a minimal level of HS5 (<5%) presenting in the MM cells FACS isolated from the lower chamber of the coculture system (data not shown). For all analyses, we excluded signature genes of stromal cells, defined as the top 1000 highest-expressed genes in HS5 [31], and at the same time, silenced in MM cells (RPKM < 2). For each MM cell line, we also excluded genes of which the chromatin at the promoter was not open (as quantified by Omni-ATAC) in all three culture conditions.

#### 2.6.2. Omni-ATAC Data Analysis

Pair-end Omni-ATAC sequence reads were aligned to hg38 using Bowtie2 [32]. Reads mapped to multiple positions of the genome were excluded, and only one read was kept if multiple reads mapped to the same position. Read distribution across the genome was visualized with IGV [33]. Read-enriched regions (peaks) were predicted by MACS3 [34]. We compiled a list of reference peaks for each MM cell line: A peak is in the reference if it is present in at least three samples. An in-house script was applied to count the number of reads within each reference peak. Differential peaks were predicted by EdgeR3 [22] (FC > 1.5 and FDR < 0.01) or GFOLD [35] (|GFOLD value| > 0.585, corresponding to a fold change of 1.5). A peak is in distal genomic regions if it does not overlap with any promoters, defined as ±2500 bps from transcription start sites from RefSeq annotation.

Target genes of differential peaks were predicted by GREAT [36]. Functional inference of the target genes was performed with GREAT on biological processes, with EnrichR [37] on Wikipathway, and webgestalt [38] on drug terms in GLAD4U [39]. To infer TF’s potential regulators, motif analysis applied to differential peaks (±50 bps from the summits) was performed with MEME (de novo) [40] or HOMER (motifs derived from public ChIP-Seq datasets [41]), with peaks not differentially accessible as the background. Detection of the presence of a motif in a genomic sequence was performed by FIMO [42]. Motifs were annotated by searching against HOCOMOCO Human (v11 CORE) [43] using TOMTOM [40].

#### 2.6.3. Overall Survival Analysis

We used expression data and survival outcomes from five MM patient cohorts to define unfavorable prognostic markers for MM: The CoMMpass clinic trial (IA14) [44], MAPQ-II (GEO: GSE24080) [45], APEX/SUMMIT (GSE9782) [46], TT2 (GSE4204) [47], and TT6 (GSE57317) [48]. For each gene *G* from each cohort, the patients were sorted into the top 25% and the remaining portion based on the expression level of *G*. The log-rank *p*-value from Kaplan–Meier analysis and the hazard ratio from the Cox proportional-hazards model were used to assess the significance of *G* being an unfavorable prognostic marker on overall survival (*p*-value < 0.05 & HR > 1). A gene was considered an unfavorable prognostic marker if it was supported by the CoMMpass clinic trial and also by at least one of the other four cohorts.

## 3. Results

### 3.1. An Overall View of BMSC-Induced Expression Changes in MM Cells

To define BM stroma-induced transcriptome signatures of MM cells, we utilized an MM-BMSC transwell coculture system and conducted RNA-Seq analysis for MM cells in the upper transwell (TSW) and MM cells in the lower chamber where adhesive BMSCs coexisted (CO); MM cells in the monoculture (MONO) served as the baseline (Figure 1A and Appendix A). We repeated the experiments with two MM cell lines (MM.1S and RPMI8226) and obtained consistent changes in expression relative to the monoculture (Appendix A). Therefore, unless specifically noted, we used MM.1S as a representative to illustrate the findings.

Principal Component Analysis (PCA) applied to samples based on gene expression profiles revealed that MM cells from the upper transwell and from the lower chamber formed a cluster separate from the monoculture (Figure 1B). Compared to the monoculture, 640 genes showed increased expression and 518 genes showed decreased expression in MM cells from the lower chamber (Figure 1C; left panel). Similar numbers of changes in gene expression were observed for MM cells from the upper transwell (Figure 1C; middle panel). A limited number of differentially expressed genes were detected between MM cells from the upper transwell and the lower chamber (Figure 1C; right panel). We visually inspected the expression patterns of genes across the three groups of MM cells using a heat map. In this analysis, we focused on genes differentially expressed in any two of the three MM populations (Figure 1C). As expected, the transcription signatures of MM cells from the upper transwell and the lower chamber were similar but distinctive to those from the monoculture (Figure 1D), with individual genes exemplified for *MCM5* and *SOCS3* (Figure 1E).

We validated the BMSC-induced transcriptome signatures by comparing our RNA-Seq data to expression datasets published by others using a similar co-culture system [30,31]. McMilin et al. [30] isolated GFP-labeled MM.1S cells through fluorescence-activated cell sorting (FACS) after a 24-h coculture with HS5 cells and profiled gene expression using the U133plus2.0 Affymetrix array. Genes upregulated in the lower chamber (CO) vs. the monoculture (MONO) from our data were also upregulated when aligned with the expression data published by McMilin et al. [30] (Appendix A; left panel), though the results for downregulated genes were less impressive (Appendix A; right panel). Lam et al. [31] separated MM.1S cells from a 24-h coculture with HS5 cells using CD138^+^ MicroBeads and profiled gene expression with RNA-Seq. Though the data were generated using an early version of the RNA-Seq protocol without replicates [31], the expression changes were generally consistent with ours for both the upregulated and the downregulated genes (Appendix A). Fagerli et al. [27] defined a set of genes induced by IL6, TNF-α, IL21, or co-culture with MM patient-derived BMSCs in several myeloma cell lines (IH-1, OH-1, and INA-6) using Affymetrix microarrays. Compared to the monoculture, cytokine-induced genes were generally upregulated for MM cells from the upper transwell (Appendix A); similar results were obtained when another two sets of IL6-induced genes were considered, one for INA-6 [28] (Appendix A) and another for ANBL-6 [29] (Appendix A).Therefore, the BMSC-induced transcriptome signatures of MM cells defined in our work were validated by gene expression data published by others [27,28,29,30,31].

### 3.2. BMSC-Induced Transcriptome Signatures Associated with Soluble Factors

BMSCs interacting with MM cells activate the secretion of growth-promoting soluble factors including IL6, IGF-1, SDF1-α, and VEGF, which in turn affect MM cells and vice versa [8]. The transwell membrane allows soluble factors to pass through, but not BMSCs. Thus, MM cells from the transwell are affected by soluble factors secreted from BMSCs but not by physical adhesion to BMSCs. The membrane also allows exosomes to pass through, and thus the soluble factors mentioned throughout the manuscript do not necessarily exclude exosomes.

We compared MMs from the upper transwell to those from the monoculture to define soluble-factor-induced transcriptome signatures. As expected [8], results from GSEA [24] against MSigDB hallmark gene sets [25] revealed that soluble factors upregulated several signaling pathways including IL6/JAK2/STAT3 with leading genes including SOCS3 and STAT3, TNFα signaling via NFκB, and an interferon γ response (Figure 2A), as well as unfolded protein response genes for MM cells (Figure 2B). MYC targets were downregulated (Figure 2C), which are positively correlated with cell proliferation [49].

BMSCs revert MM cells to a less-differentiated phenotype [50]. Plasma cells in the bone marrow differentiate from early plasma cells. We extracted signature genes of early plasma cells (vs. other B cells) from the NB2PC atlas. The expression of the signature genes was generally higher in MM cells from the upper transwell than from the monoculture (Figure 3A). An early study of myeloma-propagating activity identified a population of CD138^−^ MM cells, also known as Pre-PC cells, which emerged between plasmablasts and plasma cells [26]. The Pre-PC cells in myeloma could be generated by reverse differentiation from matured CD138^+^ PCs and are drug resistant in MM patients [26]. Microarray gene expression data from MM patients define a set of signature genes of Pre-PC as compared to plasma cells [26]. MM cells from the transwell exhibited higher expression overall than from the monoculture for the Pre-PC signature genes (Figure 3B). Thus, BMSCs induced dedifferentiation of MM cells at an early stage, at least for a subpopulation of cells in terms of transcriptome changes.

Proteasome inhibitors (PIs) are part of the standard of care for the treatment of MM patients. PIs induce unfolded protein stress in MM cells [51]. RNA-Seq gene expression data for the transcription response of MM cells to PIs are publicly available [52,53]. Interestingly, soluble factor-induced genes in MM cells were also upregulated in response to bortezomib (Appendix A; left panel). Similarly, genes downregulated by soluble factors were generally downregulated by bortezomib (Appendix A; right panel). Consistent results were obtained for another proteasome inhibitor, carfilzomib (Appendix A). In contrast to transient treatment, MM cells evolve the acquired drug resistance during prolonged and repetitive treatment with PIs with an increasing dose. By reanalyzing recent RNA-Seq data for bortezomib- or carfilzomib-resistant RPMI8226 [54], we found no significant correlation between the expression changes induced by soluble factors and those induced by chronic treatment with bortezomib (Appendix A) or carfilzomib (Appendix A). Thus, the coculture of MM cells with BMSCs recapitulated the transcription signatures of a transient treatment of PIs but not a chronic treatment, which induces drug resistance likely through genetic mutations.

### 3.3. BMSC-Induced Transcriptome Signatures Associated with Physical Adhesion

Physical adhesion to BMSCs also contributes to the de novo drug resistance developed in MM cells [3]. MM cells from the upper transwell were stimulated by soluble factors, while those from the lower chamber were adherent to BMSCs and therefore were additionally affected by physical adhesion. A direct comparison of the two defined the transcription changes attributed to adhesion. The numbers of differentially expressed genes from MM.1S cells were limited (Figure 1C, right panel), preventing thorough characterization. Therefore, we switched to the RPMI8226 cell line, where several hundred differentially expressed genes were detected (Appendix A; right panel). The expression changes induced by adhesion from our data were consistent with those from a two-channel expression microarray that measured the difference between RPMI8226 cells cultured with or without adhesion to UBE6T-7 BMSCs [5] (Appendix A). GSEA analysis revealed an upregulation in several signaling pathways, including TGFβ, TNFα via NFκB, and Kras, and downregulation in DNA replication (Appendix A). To explore the correlation between changes in expression induced by soluble factors and, additionally, those by adhesion, we applied contingency table analysis to compare the differentially expressed genes between TSW vs. MONO and CO vs. TSW. The number of genes showing a consistent change in both comparisons was over 2-fold higher than expected (Appendix A). While no specific biological process was remarkable for the shared upregulated genes, the commonly downregulated genes were linked to proliferation-related biological processes (Appendix A). Therefore, both soluble factors and adhesion contributed to a transcription program corresponding to slowed proliferation for MM cells when in a coculture with BMSCs.

### 3.4. An Overall View of BMSC-Induced Transformation of Regulome in MM Cells

The described transcriptional reprogramming of MM cells induced by BMSCs occurred within 48 h, indicating a contribution from epigenetic mechanisms rather than genetic mutations. We therefore profiled and compared accessible chromatin (regulome) for MM cells in the coculture and monoculture using Omni-ATAC (Appendix A) [17]. We repeated the experiments with two MM cell lines (MM.1S and RPMI8226) and obtained consistent changes in the landscapes of chromatin accessibility (Appendix A). Therefore, we chose MM.1S as a representative cell line to illustrate the findings, unless specifically stated otherwise.

Omni-ATAC identified ~72 K accessible regions for MM.1S, denoted as the reference peaks thereafter. PCA results based on chromatin accessibility at the reference peaks revealed that samples from the lower chamber (CO) and those from the upper transwell (TSW) were clearly separated from the monoculture (Figure 4A). Differential peak analysis revealed an increase in chromatin accessibility for 1200 to 3000 peaks and a decrease for 4000 to 6000 peaks in the presence of BMSCs as compared to the monoculture (Figure 4B; left two panels). Similar to gene expression analysis, the landscapes of accessible chromatin were highly similar between MM cells from the upper transwell and those from the lower chamber (Figure 4B; right panel). Because changes in chromatin accessibility relative to monoculture were generally consistent between the upper transwell and the lower chamber (Figure 4C), we focused on the difference between MM cells from the upper transwell and the monoculture for downstream analysis. We applied GREAT analysis [36] to infer target genes for the differential peaks. Target genes of peaks increasing in accessibility were enriched in biological processes linked to responses to external stimulus and the regulation of apoptosis (Figure 4D; left panel), while those decreasing in accessibility were enriched in functions related to chemotaxis and the positive regulation of the JAK-STAT cascade (Figure 4D; right panel). As expected, target genes associated with peaks increasing in accessibility were generally upregulated in expression when compared to other genes (Figure 4E).

### 3.5. Genes Increasing Accessibility at Multiple Regulatory Sites Are Clinically Relevant to MM

A concordant increase in gene expression and chromatin accessibility at nearby regulatory regions were observed for several genes known to be induced by BMSCs. SOCS3 is part of a negative feedback loop strongly induced by the JAK-STAT signaling pathway [55]. This gene was silenced in the monoculture, yet showed an open chromatin configuration at the promoter (Figure 5A). BMSCs activated the transcription of *SOCS3* (FDR < 2.6 × 10^×2212;^) with an increase in chromatin accessibility at the promoter and multiple nearby distal sites (Figure 5A; left panel). ABCA1 is a member of the ATP-binding cassette (ABC) transporters known for their contributions to multidrug resistance [56]. Similar to *SOCS3*, the promoter of *ABCA1* was open in the monoculture. BMSC induced an increase in accessibility at the promoter and several regulatory sites located at 13-to-125K bps upstream (Figure 5B; middle panel), consistent with transcription activation of *ABCA1* (FDR = 1.4 × 10^−10^). The *SGK1* locus encodes the serum and glucocorticoid-regulated protein kinase 1 (SGK1), known to promote cellular proliferation and apoptosis protection with a rapid induction through the JAK-STAT signaling pathway [27]. While the promoter of *SGK1* showed no remarkable change in accessibility, the expression upregulation by BMSCs (FDR = 4.0 × 10^−8^) was associated with an increase in chromatin accessibility at multiple distal regulatory sites in the nearby region (Figure 5C; right panel).

A genome-wide survey identified 232 genes that were expressed and associated with multiple genomic regions (>2) that increased in chromatin accessibility in the presence of BMSCs (Appendix A). The mRNA expression level of the 232 genes was generally upregulated by BMSCs as compared to other genes (Figure 5B). WikiPathway analysis with EnrichR [37] revealed enrichment in pathways related to angiogenesis, apoptosis regulation, and unfolded protein stress (Figure 5C). Interestingly, these genes were enriched in drug-target terms such as bortezomib and melphalan (Figure 5D), two drugs commonly used in the treatment of MM patients. We further investigated the connection of the 232 genes to the clinical outcome of MM patients in terms of overall survival. A substantial portion of the genes (21%) resulted in being unfavorable prognostic markers of overall survival in at least two MM patient cohorts (Figure 5E). A prominent example was shown for MCL1 (Figure 5F), an anti-apoptotic protein highly expressed in MM with a crucial role in disease progression [57]. A further literature review identified nine genes implicated in promoting drug resistance in MM cells, including CDK6 [58], HIF1A [59], MCL1 [60], MITF [61], PPP3CA [62], PTP4A3 [63], RUNX3 [64], SGK1 [27,65], and USP14 [66] (Appendix A).

### 3.6. Candidate Regulators of BMSC-Induced Transformation of Regulome

Changes in chromatin accessibility predominantly occurred in genomic regions distal to promoters (Figure 6A). To identify potential transcription factors (TFs) contributing to the BMSC-induced chromatin reorganization, we applied de novo motif finding [40] to the distal peaks that increased or decreased in accessibility for MM cells from the transwell relative to the monoculture. By taking peaks not differentially accessible as the background, we identified DNA motifs corresponding to JUNB and the heterodimer of ATF::CEBP for peaks showing an increase in accessibility (Figure 6B); for completeness, we also listed the motifs for peaks showing a decrease in accessibility (Appendix AA).

The proto-oncogene JUNB is an AP-1 family member known as a mediator of MM cell survival, proliferation, drug resistance, and BM angiogenesis [67,68]. Consistent with earlier observations [68], BMSCs upregulated the expression of JUNB in MM cells (Figure 6C). The JUNB motif presented at 25% of the distal peaks showing an increase in accessibility (Figure 6B); a potential enhancer site located 27K bps downstream of VEGFA served as an example (Figure 6D), also validated by ChIP-seq data in IL6-stimulated MM.1S cells [67]. At a genome-wide scale, reference peaks overlapped with JUNB binding sites defined by ChIP-seq data [67] exhibited an overall increase in chromatin accessibility for MM cells in the transwell vs. the monoculture (Appendix A). Intriguingly, GREAT-predicted target genes of JUNB-motif-containing distal peaks increased their expression in the upper transwell as relative to the monoculture (Figure 6E). Gene ontology enrichment analysis identified responses to drugs as the top hit (Figure 6F). WikiPathway analysis revealed that the target genes were enriched in the chemokine signaling pathway and the VEGFA-VEGFR2 signaling pathway (Appendix A), consistent with the role of JUNB in regulating MM BM angiogenesis [67].

MM cells constantly maintain high demand in the ubiquitin-mediated proteasomal degradation of misfolded proteins, which activates an integrated stress response. ATF4 activation is essential to alleviate the stress for cell-intrinsic metabolic adaptations [69]. In addition to the JUNB motif, two motifs corresponding to a heterodimer of ATF::CEBP were discovered in distal peaks showing an increase in accessibility (TSW vs. MONO) (Figure 6B). The ATF::CEBP motifs presented in 7–8% of the distal peaks, showing an increase in accessibility (Figure 6B). A specific example was shown for a putative enhancer located 10K bps downstream from PIM3 (Figure 6G), a Ca(2+) /calmodulin-dependent protein kinase known to prevent apoptosis and promote cell survival [70]. GREAT-predicted target genes of the chimeric motifs were generally upregulated in the upper transwell vs. the monoculture (Figure 6H). Gene ontology enrichment analysis identified biological processes associated with the regulation of cell death, cell–cell adhesion, and the response to drugs (Figure 6I). WikiPathway analysis revealed that the target genes were enriched in the IL6 signaling pathway (Appendix A). The results suggested that members from the heterodimer of ATF::CEBP, similarly to JUNB, contributed to the chromatin reorganization linked to the transcription response to BMSCs.

The de novo motif analysis did not recover a motif for STAT3, which is a downstream TF activated by IL6. Nevertheless, scanning the motifs derived through ChIP-seq data by HOMER [41] in distal genomic regions that showed an increase in accessibility (TSW vs. MONO) identified a STAT3 motif in addition to the JUNB and ATF::CEBP motifs (Appendix A). Interestingly, the STAT3 motif presented in only 2.83% of the genomics regions, in contrast to 27.4% for JUNB and 5.0% for ATF::CEBP (Appendix A). Transcription regulators can also be inferred with a set of genes through an association analysis with targets predicted from public ChIP-seq data using TFEA.ChIP [23]. We applied the analysis to genes that were upregulated in the transwell as compared to the monoculture. The analysis revealed STAT3 as one of the top transcription regulators (Appendix A). Therefore, considering only modest enrichment of the STAT3 motif was observed on differential peaks, the BMSC-activated STAT3 likely acts on pre-existing accessible chromatin for transcription regulation.

## 4. Discussion

Multiple myeloma (MM) is a disease of malignant plasma cells predominantly growing in the bone marrow (BM). As important components of the BM microenvironment, BM stromal cells (BMSCs) interact with MM cells and contribute to the development of drug resistance [71]. Specifically, soluble factors secreted from BMSCs and physical adhesion to BMSCs activate a panel of signaling pathways in MM cells, resulting in transcriptome reprogramming leading to drug resistance [8,30]. In contrast to the acquired drug resistance, BMSC-induced drug resistance is rapidly triggered by signaling events, which are reversible and not likely caused by genetic mutations [2,3]. Presumably, effector TFs downstream of activated signaling pathways assume roles in regulating transcriptome reprogramming. However, whether the transcription regulation acts on a pre-existing chromatin configuration or involves an epigenetic re-configuration remains unclear. To address this question, we employed an in vitro coculture system of MM cells with BMSCs and the transcriptome and regulome for MM cells affected by soluble factors and/or physical adhesion to BMSCs or in a monoculture. This resource allowed us to examine BMSC-induced chromatin reorganization in MM, its connection to transcription reprogramming linked to its drug response, and potential regulators contributing to the chromatin reorganization.

Previous gene expression analysis based on a microarray [27,28,29,30] or an early version of the RNA-Seq protocol [31] identified transcription signatures of MM cells interacting with BMSCs. The signatures were generally recapitulated by our study with the most recent RNA-Seq technique. Moreover, we employed a transwell coculture system to compare MM cells in the upper transwell, MM cells adherent to BMSCs in the lower chamber, and MM cells in monoculture to delineate the contribution from soluble factors and physical adhesion. While substantial changes in the transcriptome were observed when compared to the monoculture, the difference between MM cells in the transwell and those in the lower chamber was marginal. Thus, soluble factors other than physical interaction accounted for most of the BMSC-induced changes in the transcriptome. We reasoned that soluble factor-mediated drug resistance is largely mediated by transcription induction, whereas cell-adhesion-mediated drug resistance is more likely mediated by non-transcriptional mechanisms.

We have characterized the transcriptome signatures induced by BMSCs: Soluble factors from a coculture of MM and BMSCs upregulated gene targets of signaling pathways related to cell proliferation and drug resistance, including JAK2/STAT3, PI3K/AKT, Ras, and NF-κB [8]. The results are consistent with previous studies based on MM cells in a coculture [30,31] or cytokine-stimulated MM cells [27,28,29]. BMSCs also upregulated genes involved in the unfolded protein response and recapitulated transcription response to PIs, consistent with other observations that MM cells interacting with BMSCs become less sensitive to bortezomib [72,73]. Lastly, soluble factors upregulated the expression of signature genes for early plasma cells and resistant Pre-PCs in ND MM patients [26], corresponding to normal and malignant plasma cells, respectively. The results supported a previous view that MM cells interacting with BMSCs exhibit a less-differentiated and drug-resistant phenotype [50].

Chromatin consists of the genomic DNA sequence wrapping around nucleosomes. The landscape of accessible chromatin changes dynamically in response to both external stimuli and developmental cues [74]. Evaluating changes in the regulome emerges as a powerful tool to decode regulatory mechanisms contributing to myelomagenesis and drug resistance [75,76,77,78,79]. We examined the BMSC-induced transformation of the regulome and its connection to expression changes in genes linked to drug responses. The difference in the regulome for MM cells in the upper transwell and those in the lower chamber was minimal, confirming soluble factors as the major source in inducing transcription changes. In contrast, MM cells in the transwell exhibited substantial changes in the regulome as compared to the monoculture. In particular, we identified a subset of genes where BMSCs induced an increase in chromatin accessibility at multiple regulatory sites, including genes known to promote drug resistance such as ABCA1 [56], SGK1 [27], and MCL1 [57]. These signature genes were enriched in biological processes related to IL6 signaling, apoptosis, angiogenesis, unfolded protein stress, and the transcription response to drugs. Clinically relevant is their enrichment in unfavorable prognosis markers on overall survival. The BMSC-induced increase in chromatin accessibility was coupled with an overall upregulation of expression. Thus, the BMSC-induced transformations of the regulome and transcriptome are coupled for genes linked to the drug response and clinically relevant to MM.

The profile of accessible chromatin, when combined with motif-finding or foot-printing analysis, helps to infer TF regulators of the chromatin landscape [80,81,82]. JUNB was the most prominent TF enriched in distal genomic regions that gained accessibility as induced by BMSCs. Compared to other AP-1 family TFs, JUNB in MM cells is specifically induced by BMSCs [68]. Predicted gene targets of distal sites containing the JUNB motifs and increasing in accessibility were enriched in the VEGFA-VEGFR2 signaling pathway and the response to drugs, consistent with its roles in angiogenesis and drug resistance in MM cells [67,68]. Our results supported JUNB in promoting chromatin accessibility, a role validated in other cell systems, including cardiac regeneration [83], hormone stimuli [84], and T-cell activation [85]. Potential mechanisms may include the SWI/SNF (BAF) chromatin remodeling complex recruited by AP-1 family members such as JUNB to establish accessible chromatin [86,87]. The next two prominent motifs corresponded to the heterodimer of ATF::CEBP. The attractive TF pair included ATF4::CEBPβ [88,89,90], both of which were upregulated in the upper transwell relative to the monoculture (Figure 6C). ATF4 becomes activated in response to an integrated stress response [69] or pro-growth signaling [91]. While basal expression of ATF4 may attenuate PI-induced apoptosis in MM cells [92], ATF4 induction promotes PI resistance by facilitating lipogenesis [93]. In supporting a role in regulating chromatin accessibility, ATF4 promotes histone acetylation [94], reduces repressive H3K9 methylation in target genes [95], and is implicated as a pioneer factor by recognizing methylated DNA together with CEBPβ [90]. As a partner of ATF4, CEBPβ activates TFs critical for MM proliferation [96] and mediates resistance to IMiD compounds [97]. As a recipient of extracellular signals [98], CEBPβ acts as a pioneer factor in remodeling chromatin for transcription regulation [99]. Together with our results, we speculated that JUNB, ATF4, and CEBPβ positively contribute to the BMSC-induced transcriptome changes in MM cells by facilitating the transformation of chromatin accessibility

## 5. Conclusions

We characterized BMSC-induced transformations of the transcriptome and regulome in MM cells, which were most likely driven by soluble factors. Genes with an increase in chromatin accessibility at multiple regulatory sites served as prominent targets for further investigation into their functional connection to drug response and their overall adverse clinical prognosis in MM. As for mechanisms, we identified candidate regulators contributing to the transformation of regulome coupled with changes in the transcriptome. Future work should include loss-or gain-of-function assays, ChIP-seq assays, and a reevaluation of the transformation to depict the epigenetic roles of the proposed TFs in regulating the transcription program underlying BMSC-induced drug resistance.

## Figures and Tables

**Figure 1 cancers-14-00927-f001:**
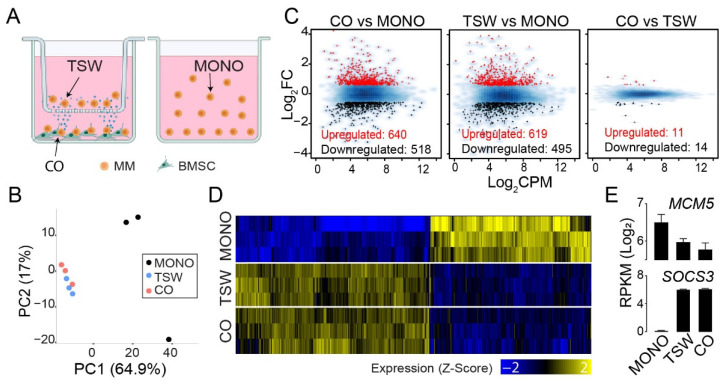
BMSC-induced transcriptome signatures of MM cells. (**A**) The co-culture model of MM cells with BMSCs using a transwell. TSW: MM cells in the upper transwell; CO: MM cells from the lower chamber where adhesive BMSCs coexisted; MONO: MM cells in monoculture. (**B**) PCA applied to samples from MONO, TSW, and CO based on gene expression values. (**C**) MA-plots displaying Log_2_CPM (X-axis) and fold change of expression (y-axis) for the comparisons of CO vs. MONO (left), TSW vs. MONO (middle), and CO vs. TSW (right). Red: Genes upregulated in expression. Black: Genes downregulated. Blue: All expressed genes. (**D**) Heat map visualization of expression level for genes (columns) differentially expressed between any two from MONO, TSW, and CO. Expression values were normalized into z-scores for each gene across all samples. (**E**) Comparisons of gene expression level (RPKM; log_2_) across MONO, TSW, and CO for *MCM5* and *SOCS3*. Error bar: Standard deviation.

**Figure 2 cancers-14-00927-f002:**
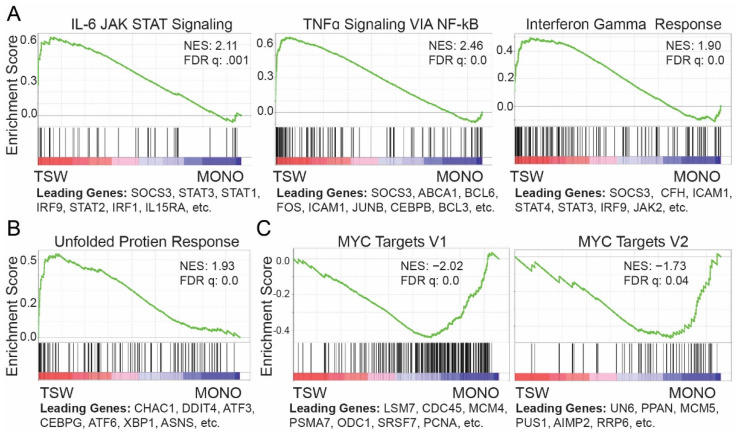
BMSCs affected the expression of several MSigDB hallmark gene sets in MM cells. (**A**–**C**) GSEA of expressed genes sorted by fold change of expression (TSW/MONO) from high (red) to low (blue) against MSigDB hallmark gene set (vertical bars): IL6/JAK/STAT signaling (**A**, left), TNFα signaling via NFκB (**A**, middle), INFγ response (**A**, right), unfolded protein response (**B**), MYC targets V1 (**C**, left), and MYC targets V2 (**C**, right).

**Figure 3 cancers-14-00927-f003:**
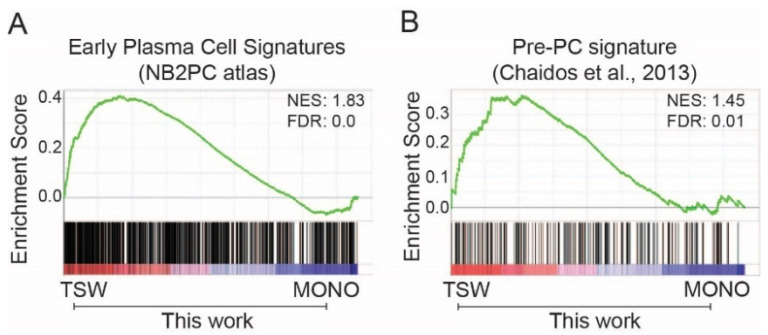
BMSC-coculture recapitulated gene signatures of early B cells in MM. (**A**,**B**) GSEA of expressed genes sorted by fold change of expression (TSW/MONO) from high (red) to low (blue) against signature genes (vertical bars) of early plasma cells from the NB2PC atlas (**A**) or signature genes of Pre-PC from newly diagnosed (ND) MM patients (**B**), see [26].

**Figure 4 cancers-14-00927-f004:**
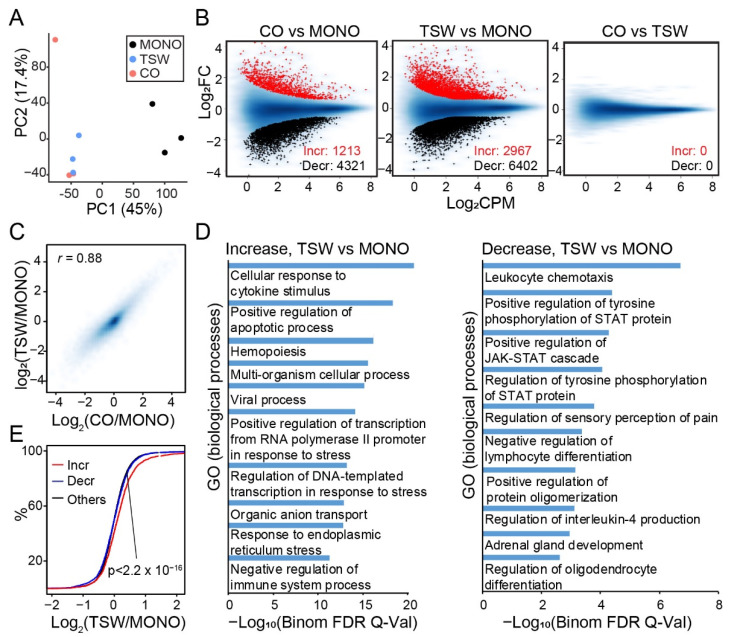
BMSCs induced substantial transformation of regulome in MM. (**A**) PCA applied to samples from MONO, TSW, and CO based on chromatin accessibility at reference accessible regions (peaks) defined by Omni-ATAC. (**B**) MA-plots displaying Log_2_CPM (X-axis) and fold change of chromatin accessibility (Y-axis) for the comparisons of CO vs. MONO (left), TSW vs. MONO (middle), and CO vs. TSW (right). Red: Reference peaks increasing in accessibility (Incr); Red: Reference peaks decreasing in accessibility (Decr); blue: All reference peaks. (**C**) Smoothed scatter plot comparing the fold changes in accessibility at reference peaks between CO vs. MONO and TSW vs. MONO. *r*: Pearson correlation coefficient. (**D**) Top ten enriched gene ontology (GO) terms on biological processes for GREAT-predicted target genes of reference peaks that increased (left panel) or decreased (right panel) in accessibility from MONO to TSW. (**E**) Empirical cumulative distribution of the fold change of expression (TSW/MONO) of target genes predicted for peaks showing increase (Incr), decrease (Decr), or no change (Others) in accessibility from MONO to TSW. A line shifting to the right indicates an overall increase in the fold change of expression. *p*-value by the Kolmogorov–Smirnov (K-S) test.

**Figure 5 cancers-14-00927-f005:**
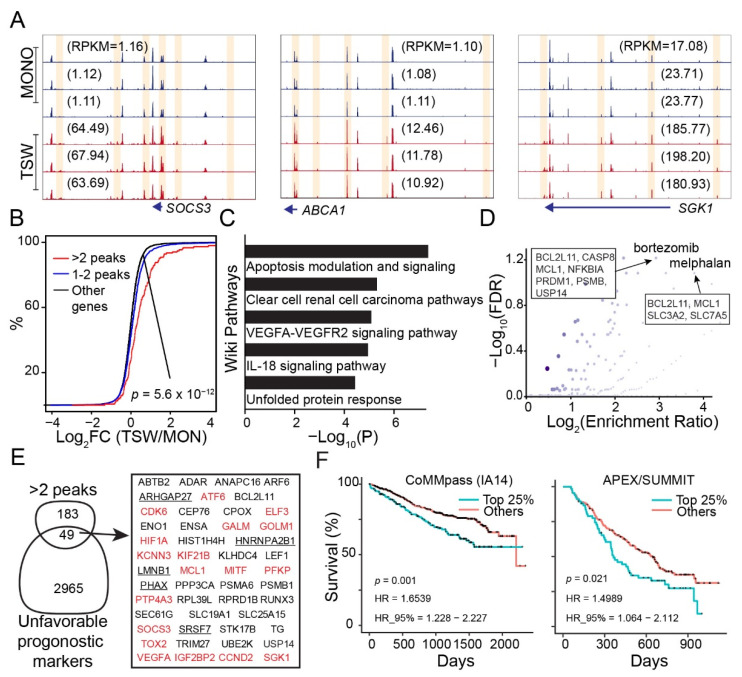
Genes with multiple peaks that increased in chromatin accessibility as induced by BMSCs are clinically relevant to MM. (**A**) Omni-ATAC read distribution around the genes *SOCS3* (left), *ABCA1* (middle), and *SGK1* (right) across MM samples from the monoculture (MONO) and the upper transwell (TSW). Y-axis (normalized read density) adjusted to the same scale within each plot. Highlighted in yellow: Peaks increasing in accessibility from MONO to TSW; numbers in parenthesis: Gene expression values in RPKM (decimal). (**B**) Empirical cumulative distribution of the fold change of expression (TSW/MONO) of genes associated with 1–2 peaks or multiple peaks (>2) that increased in accessibility from MONO to TSW. *p*-value by the K-S test. (**C**) Gene pathways enriched for the genes associated with multiple peaks as in panel (**B**). (**D**) Drug-based enrichment analysis based on the GLAD4U database [39] for the genes associated with multiple peaks as in panel (**B**). Y-axis: The significance of overlapping by FDR. X-axis: Enrichment ratio (observed number/expected number); blue color with intensity proportional to the number of overlapped genes. (**E**) Venn diagram for genes associated with multiple peaks as in panel (**B**) and unfavorable prognostic markers in overall survival as defined from several MM patient cohorts (see Section 2). Red: Expression upregulated in TSW vs. MONO (FDR < 0.1). Underlined: Expression downregulated in TSW vs. MONO (FDR < 0.1). (**F**) Kaplan–Meier survival plot for MCL1 from the CoMMpass trial (left) or the APEX/SUMMIT trial (right). Patients sorted into the top 25% and others based on MCL1 expression. *p*-value by log-rank test.

**Figure 6 cancers-14-00927-f006:**
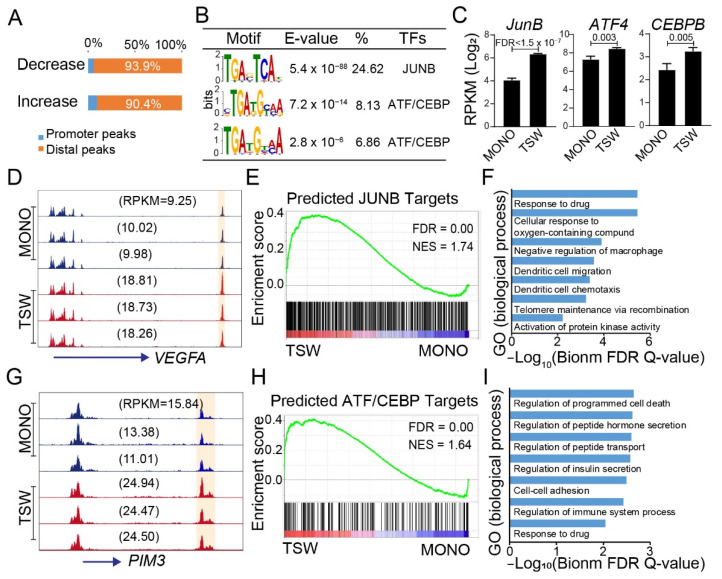
TFs predicted to regulate BMSC-induced transformation of regulome. (**A**) Distribution of differential peaks in promoter regions and distal non-promoter regions grouped by their changes from monoculture (MONO) to the upper transwell (TSW). (**B**) Top motifs discovered by MEME for peaks increasing in accessibility from MONO to TSW, with E-value, % of peaks containing the motif, and representative TFs indicated. (**C**) Comparisons of gene expression level (RPKM; log_2_) between MONO and TSW for *JUNB, ATF4,* and *CEBPB*. Error bar: Standard deviation. FDR calculated by EdgeR. (**D**) Omni-ATAC read distribution around *VEGFA* across MM samples from MONO and TSW. Y-axis (normalized read density) adjusted to the same scale. Highlighted in yellow: Peaks containing JUNB motif; numbers in parenthesis: Gene expression level in RPKM (decimal). (**E**) GSEA of expressed genes sorted by fold change of expression (TSW/MONO) from high (red) to low (blue) against genes (vertical bars) predicted as targets of peaks that increased accessibility and contained JUNB motif(s). (**F**) Biological processes enriched with predicted JUNB targets as in panel (**E**). (**G**) Similar to panel (**D**) but for *PIM3*; highlighted in yellow: Peaks containing ATF::CEBP motif (**H**) Similar to panel (**E**) but for targets of peaks that increased accessibility and contained ATF::CEBP motif(s). (**I**) Biological processes enriched for predicted ATF::CEBP targets as in panel (**H**).

## Data Availability

Sequencing data for RNA-Seq and Omni-ATAC are accessible from the Gene Expression Omnibus (GEO) with accession number GSE193658.

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
