# Peer review of "Bone Marrow Stroma-Induced Transcriptome and Regulome Signatures of Multiple Myeloma"

_cancers, 2022, doi:10.3390/cancers14040927_

Round 1

Reviewer 1 Report

A well-designed and well-performed study on effects of interactions of bone marrow stem cells (BMSC) cell line on transcriptome and chromatin accessibility of two multiple myeloma (MM) cell lines, with adequate numbers of experimental replicates and the results concordant between the two MM cell lines. The interaction between MM and BMSC was studied before at MM transcription level, as duly indicated by the Authors. The novelty of this study lies in the use of transwell coculture system, which permitted comparison between the effects of secreted factors and direct cell-cell interactions, and in the quantitative analysis of MM chromatin accessibility. The study demonstrates that the effects of BMSC on MM are largely mediated by soluble factors; that transcriptional regulation generally occurs in open promoter context, while the changes in chromatin accessibility largely in promoter-distal regions. The genes that show transcriptional induction and increased chromatin accessibility are more strongly induced transcriptionally, enriched in drug-target genes, with many of them predictive of MM patients survival. By motif analysis of chromatin regions of increased accessibility, the Authors identified JUND and ATF4::CEBP heterodimers as likely regulators of the chromatin accessibility in MM cells, while STAT3 was identified as likely transcriptional activator in this system. A very valuable part of this study is comparison of the BMSC-induced changes in MM transcriptome to transcriptional changes in B-cell differentiation, and changes in the MM transcriptome in response to cytokines and drugs, in particular the proteasome inhibitors.

This manuscript is acceptable for publication in its current form.

The Authors may choose to follow or ignore these two minor remarks:

The MEME background set is described as “peaks showing no change in accessibility” in Materials and Methods, and as “peaks with no remarkable change change in accessibility”. The Authors may wish to clarify the filtering criteria.

The rationale for the conclusion in the sentence: “Therefore, while JUNB and ATF::CEBP TFs are implied in regulating chromatin reconfiguration at distal sites, the BMSC-activated STAT3 more likely acts on pre-existing accessible chromatin for transcription regulation.” (lines 474-476) only becomes clear in the Discussion, after the known pioneer factor role of the former 3 TFs, but not of STAT3, is recalled. Consider moving this sentence to line 559 after ref. 90.

Author Response

A well-designed and well-performed study on effects of interactions of bone marrow stem cells
(BMSC) cell line on transcriptome and chromatin accessibility of two multiple myeloma (MM) cell
lines, with adequate numbers of experimental replicates and the results concordant between the
two MM cell lines. The interaction between MM and BMSC was studied before at MM transcription
level, as duly indicated by the Authors. The novelty of this study lies in the use of transwell
coculture system, which permitted comparison between the effects of secreted factors and direct
cell-cell interactions, and in the quantitative analysis of MM chromatin accessibility. The study
demonstrates that the effects of BMSC on MM are largely mediated by soluble factors; that
transcriptional regulation generally occurs in open promoter context, while the changes in
chromatin accessibility largely in promoter-distal regions. The genes that show transcriptional
induction and increased chromatin accessibility are more strongly induced transcriptionally,
enriched in drug-target genes, with many of them predictive of MM patients survival. By motif
analysis of chromatin regions of increased accessibility, the Authors identified JUND and
ATF4::CEBP heterodimers as likely regulators of the chromatin accessibility in MM cells, while
STAT3 was identified as likely transcriptional activator in this system. A very valuable part of this
study is comparison of the BMSC-induced changes in MM transcriptome to transcriptional
changes in B-cell differentiation, and changes in the MM transcriptome in response to cytokines
and drugs, in particular the proteasome inhibitors.
This manuscript is acceptable for publication in its current form.

Re: We thank the reviewer for a high appreciation on the novelties of the work.

The Authors may choose to follow or ignore these two minor remarks:
The MEME background set is described as “peaks showing no change in accessibility” in
Materials and Methods, and as “peaks with no remarkable change in accessibility”. The Authors
may wish to clarify the filtering criteria.

Re: We have changed both to “peaks not differentially accessible”, to be consistent and to minimize
confusion. Thanks for the nice catching.

The rationale for the conclusion in the sentence: “Therefore, while JUNB and ATF::CEBP TFs are
implied in regulating chromatin reconfiguration at distal sites, the BMSC-activated STAT3 more
likely acts on pre-existing accessible chromatin for transcription regulation.” (lines 474-476) only
becomes clear in the Discussion, after the known pioneer factor role of the former 3 TFs, but not
of STAT3, is recalled. Consider moving this sentence to line 559 after ref. 90.

Re: We have changed the sentence to “Therefore, considering only a modest enrichment of STAT3 motif
was observed on differential peaks, the BMSC-activated STAT3 likely acts on pre-existing accessible
chromatin for transcription regulation.”

Reviewer 2 Report

The authors choose an exceptionally good modal to study BMSC-induced multi-drug resistance in MM. I have a few queries about this manuscript that I mention below.

  1. Could you also add the novelty of your study with this sentence?

The proposed roles of JUNB, ATF4, and CEBPβ in facilitating BMSC-induced chromatin transformation in MM cells are consistent with observations documented in other systems.

  1. Multi-drug resistance takes a long time before developing any type of resistance except mutation, how does this abbreviated time study replicate this phenomenon?
  2. In vivo system has multiple cells and tissue that react with bone marrow cells, and it is also one of the methods for treatment like bone marrow transfer for many diseases. So how can study between a few cells be comparable for patient cells and their treatment-related options?

Author Response

Reviewer #2:

We thanks the reviewer for his/her overall high rating on the score sheet of the manuscript.

1. Could you also add the novelty of your study with this sentence?
The proposed roles of JUNB, ATF4, and CEBPβ in facilitating BMSC-induced chromatin
transformation in MM cells are consistent with observations documented in other systems.

Re: To highlight the novelty of our work, we changed the sentence to “Together with our results, we
speculated that JUNB, ATF4, and CEBPβ positively contribute to the BMSC-induced transcriptome
changes in MM cells by facilitating the transformation of chromatin accessibility”.

2. Multi-drug resistance takes a long time before developing any type of resistance except
mutation, how does this abbreviated time study replicate this phenomenon?

Re: Several investigators have shown that MM cells exposed to components of the bone marrow
microenvironment is sufficient to confer a multi-drug resistant phenotype [1-4]. As one example, MM.1S
cells gain resistance to doxorubicin and dexamethasone within 48 hours of coculture with HS5 stromal
cells [5]. As another example, resistance of MM cells to Bortezomib occurs within 48 hours in coculture
with HS5 [6]. This type of resistance has been coined in the literature as de novo resistance associated
with the tumor microenvironment [1-4]. Acquired resistance which occurs after chronic selection of a
drug has also been shown to result in a multi-drug resistant phenotype. We agree with the reviewer that
multi-drug resistance that emerges following drug exposure can take months and in some cases is the
result of selection of multi-drug transporters such as MDR1 or BCRP which will confer multi-drug
resistance. We apologize for any confusion in the text and have added clarity in the manuscript.

3. In vivo system has multiple cells and tissue that react with bone marrow cells, and it is also one
of the methods for treatment like bone marrow transfer for many diseases. So how can study
between a few cells be comparable for patient cells and their treatment-related options?

Re: We agree with the reviewer that in vitro coculture system (this work and others [5,7-10]) is a
simplified view of the highly sophisticated bone marrow microenvironments where MM cells proliferate.
But it offers a straightforward way to investigate the impact of a single component of the BM
microenvironment on MM cells. We have identified JUNB, ATF4, and CEBPB as candidate TFs
contributing to the BMSC-induced changes in transcriptome and regulome underlying de novo drug
resistance. Our further work will generate JUNB or CEBPB depleted MM cell lines and examine their roles
in MM engraftment and response to drug treatment in mouse models. These in vivo studies will help
transform the findings from our in vitro study a step closer to validate these TFs as drug targets.

References
1. Meads, M.B.; Hazlehurst, L.A.; Dalton, W.S. The bone marrow microenvironment as a tumor
sanctuary and contributor to drug resistance. Clin Cancer Res 2008, 14, 2519-2526,
doi:10.1158/1078-0432.CCR-07-2223.
2. Kikuchi, J.; Koyama, D.; Wada, T.; Izumi, T.; Hofgaard, P.O.; Bogen, B.; Furukawa, Y.
Phosphorylation-mediated EZH2 inactivation promotes drug resistance in multiple myeloma. J
Clin Invest 2015, 125, 4375-4390, doi:10.1172/JCI80325.
3. Chen, W.C.; Hu, G.; Hazlehurst, L.A. Contribution of the bone marrow stromal cells in mediating
drug resistance in hematopoietic tumors. Curr Opin Pharmacol 2020, 54, 36-43,
doi:10.1016/j.coph.2020.08.006.
4. Hideshima, T.; Mitsiades, C.; Tonon, G.; Richardson, P.G.; Anderson, K.C. Understanding multiple
myeloma pathogenesis in the bone marrow to identify new therapeutic targets. Nat Rev Cancer
2007, 7, 585-598, doi:10.1038/nrc2189.
5. McMillin, D.W.; Delmore, J.; Weisberg, E.; Negri, J.M.; Geer, D.C.; Klippel, S.; Mitsiades, N.;
Schlossman, R.L.; Munshi, N.C.; Kung, A.L.; et al. Tumor cell-specific bioluminescence platform to
identify stroma-induced changes to anticancer drug activity. Nat Med 2010, 16, 483-489,
doi:10.1038/nm.2112.
6. de la Puente, P.; Quan, N.; Hoo, R.S.; Muz, B.; Gilson, R.C.; Luderer, M.; King, J.; Achilefu, S.;
Salama, N.N.; Vij, R.; et al. Newly established myeloma-derived stromal cell line MSP-1 supports
multiple myeloma proliferation, migration, and adhesion and induces drug resistance more than
normal-derived stroma. Haematologica 2016, 101, e307-311,
doi:10.3324/haematol.2016.142190.
7. Fagerli, U.M.; Ullrich, K.; Stuhmer, T.; Holien, T.; Kochert, K.; Holt, R.U.; Bruland, O.; Chatterjee,
M.; Nogai, H.; Lenz, G.; et al. Serum/glucocorticoid-regulated kinase 1 (SGK1) is a prominent
target gene of the transcriptional response to cytokines in multiple myeloma and supports the
growth of myeloma cells. Oncogene 2011, 30, 3198-3206, doi:10.1038/onc.2011.79.
8. Croonquist, P.A.; Linden, M.A.; Zhao, F.; Van Ness, B.G. Gene profiling of a myeloma cell line
reveals similarities and unique signatures among IL-6 response, N-ras-activating mutations, and
coculture with bone marrow stromal cells. Blood 2003, 102, 2581-2592, doi:10.1182/blood-
2003-04-1227.
9. Brocke-Heidrich, K.; Kretzschmar, A.K.; Pfeifer, G.; Henze, C.; Loffler, D.; Koczan, D.; Thiesen, H.J.;
Burger, R.; Gramatzki, M.; Horn, F. Interleukin-6-dependent gene expression profiles in multiple
myeloma INA-6 cells reveal a Bcl-2 family-independent survival pathway closely associated with
Stat3 activation. Blood 2004, 103, 242-251, doi:10.1182/blood-2003-04-1048.
10. Lam, C.; Ferguson, I.D.; Mariano, M.C.; Lin, Y.T.; Murnane, M.; Liu, H.; Smith, G.A.; Wong, S.W.;
Taunton, J.; Liu, J.O.; et al. Repurposing tofacitinib as an anti-myeloma therapeutic to reverse
growth-promoting effects of the bone marrow microenvironment. Haematologica 2018, 103,
1218-1228, doi:10.3324/haematol.2017.174482.

Reviewer 3 Report

In this manuscript, Dziadowicz et al. evaluated bone marrow stroma induced changes in transcriptome and reguolome in Multiple Myeloma cell lines. Authors used transwell co-culture system to distinguish the impact of bone marrow stroma derived soluble factors from physical adhesion to stroma. They report no specific changes either in transcriptome or in regulome when comparing MM cells exposed to condition media or co-cultured with stroma. Overall, the manuscript is written well and experiments are all well controlled. However, most of the observations made from the study are already reported and the advances are incremental. In addition, the authors performed ATAC-Seq to investigate regulatory networks involved in stroma-induced changes in transcriptome and elucidated a role for JUNB and ATF4/CEBPB transcription factors. However, no follow up studies were done to validate a role for these transcription factors. While this could be beyond the scope of the study, given that most of the observations made in this study are reported earlier, validation work would have added strength to the manuscript.

My specific critique:

Authors observed changes unique to the physical association between MM cells and stroma in RPMI8226 cell line (N=2 per condition!) but not in MM1S line. Authors do not explain why there is such difference between the lines. In one statement the authors mention, “soluble factors mediated drug resistance is mediated by transcription induction whereas cell adhesion mediated drug resistance is via non-transcriptional mechanisms”. Does this statement applicable to RPMI8226 cell line as well?

While there were no significant differences in transcriptome (Figure 1C) observed when comparing CO vs MONO and TSW vs MONO, significant changes were presented in regulome (Figure 4B, 1213 vs 2967 and 4321 vs 6402). How many of these sites are overlapping and how many are unique to each condition? Could authors make any observations based on changes in accessibility that is unique to each condition?

Figure 4D is cropped inappropriately.

Figure 5A, RPKM values in parenthesis are slightly misleading, as one would imagine they correspond to peak height.

Figure 5D: Could authors provide a better explanation for Figure 5D?

Figure 5E: The genes that are highlighted in black are differentially accessible and are prognostic. Could authors speculate the importance of these genes in stroma-induced drug resistance?

Authors should include data showing the induction of JUNB, ATF4/CEBPB upon culture with condition medium in the main manuscript.

Genomic binding site data for JUNB with or without co-culture would have strengthened the manuscript quality.
